# TRIM Proteins in Colorectal Cancer: TRIM8 as a Promising Therapeutic Target in Chemo Resistance

**DOI:** 10.3390/biomedicines9030241

**Published:** 2021-02-27

**Authors:** Flaviana Marzano, Mariano Francesco Caratozzolo, Graziano Pesole, Elisabetta Sbisà, Apollonia Tullo

**Affiliations:** 1Institute of Biomembranes, Bioenergetics and Molecular Biotechnologies, National Research Council, CNR, 70126 Bari, Italy; f.marzano@ibiom.cnr.it (F.M.); mf.caratozzolo@ibiom.cnr.it (M.F.C.); g.pesole@ibiom.cnr.it (G.P.); 2Department of Biosciences, Biotechnology and Biopharmaceutics, University of Bari, “Aldo Moro”, 70125 Bari, Italy; 3Institute for Biomedical Technologies, National Research Council, CNR, 70126 Bari, Italy; elisabetta.sbisa@ba.itb.cnr.it

**Keywords:** CRC, chemoresistance, TRIM8, miR-17-5p

## Abstract

Colorectal cancer (CRC) represents one of the most widespread forms of cancer in the population and, as all malignant tumors, often develops resistance to chemotherapies with consequent tumor growth and spreading leading to the patient’s premature death. For this reason, a great challenge is to identify new therapeutic targets, able to restore the drugs sensitivity of cancer cells. In this review, we discuss the role of TRIpartite Motifs (TRIM) proteins in cancers and in CRC chemoresistance, focusing on the tumor-suppressor role of TRIM8 protein in the reactivation of the CRC cells sensitivity to drugs currently used in the clinical practice. Since the restoration of TRIM8 protein levels in CRC cells recovers chemotherapy response, it may represent a new promising therapeutic target in the treatment of CRC.

## 1. The CRC Therapy

Colorectal cancer (CRC) is one of the most prevalent malignancy tumors with high morbidity and mortality. Risk factors for the occurrence of CRC are related to both external factors such as diet, obesity, smoking, old age, chronic intestinal inflammation and genetic factors. The majority of CRC (70–80%) is sporadic, while around 20–30% of CRC has a hereditary component, such as Lynch Syndrome (LS) (3–4%) and the familial adenomatous polyposis (FAP) (∼1%) [1]. A high percentage of sporadic CRC is characterized by deletions, translocations and other chromosomal rearrangements identified as chromosomal instability (CIN) [2]. A smaller percentage of sporadic CRC show a defective DNA mismatch repair (MMR), caused by hypermutated regions and microsatellite instability (MSI) [3].

Moreover, CpG island methylation phenotype (CIMP), is an epigenetic cause of CRC, as it induces silencing of a range of tumor suppressor genes, including MutL Homolog 1 (*MLH1*), and one of the *MMR* genes [4,5]. Recently, a classification of CRC into four Consensus Molecular Subtypes (CMS) has been reported in the literature: CMS1 (MSI Immune, 14%), hypermutated, microsatellite unstable, strong immune activation; CMS2 (Canonical, 37%), epithelial, chromosomally unstable, marked WNT and MYC signaling activation; CMS3 (Metabolic, 13%), epithelial, evident metabolic dysregulation; and CMS4 (Mesenchymal, 23%), prominent transforming growth factor β activation, stromal invasion, and angiogenesis [6].

All of the studies carried out so far have demonstrated that an earlier diagnosis is correlated with a better prognosis [7,8]. Current treatments used for CRC include some combination of surgery, radio-/chemotherapies and targeted therapy [8]. Unfortunately, despite some advances in pharmacological therapies, the 5-year survival rate of patients with late stage CRC is very poor because of recurrence and metastasis; moreover, one essential reason for treatment failure is the presence of innate or acquired resistance, which affects 90% of patients with metastatic cancer [7,8,9]. 

The main therapy for CRC patients has been, since the 1950s, chemotherapy based on 5-fluorouracil (5-FU) [8,9,10]. This drug inhibits DNA replication, replacing thymidine with fluorinated nucleotides into the DNA, hereby causing cell death. The active metabolite of 5-FU, fluorodeoxyuridine mono-phosphate (FdUMP), inhibits thymidylate synthase (TS), the enzyme essential for the conversion of deoxyuridine mono-phosphate to deoxythymidine monophosphate in the DNA synthesis pathway [11]. Different studies show that high TS levels are closely associated with the 5-FU resistance of cancer cells [12,13]. It follows that 5-FU resistance is closely related to the expression of thymidylate synthase (TS) and patients with low TS expression show a better prognosis [14,15]. 

Other enzymes involved in the metabolism and degradation of 5-FU, such as Thymidine phosphorylase (TP), uridine phosphorylase (UP), orotate phosphoribosyl transferase (OPRT) and dihydropyrimidine dehydrogenase (DPD), are correlated with sensitivity of CRC cells to 5-FU. It is reported that higher levels of TP, UP and OPRT displayed enhanced sensitivity to 5-FU therapy [16,17,18], by contrast, DPD expression level is inversely correlated with chemosensitivity [17]. 

Capecitabine was the first oral chemotherapy drug for CRC. Thymidylate synthase (TS), is the enzyme that converts capecitabine to 5-FU and for this reason loss of function of this enzyme confers the resistance of Capecitabine [19,20]. 

Moreover, the literature reports that changes in the status of *p53* affects the sensitivity to TS inhibitors, suggesting that analysis of the status of *p53* (e.g., wild type or mutant and functionally active or not) could be useful to predict the clinical outcome of the chemotherapy with TS inhibitors [21]. 

FOLFOX is the first combined chemotherapeutic strategy which integrates the use of 5-FU with Leucovorin and Oxaliplatin (a platinum-based chemotherapeutic drug approved for the treatment of CRC).

Oxaliplatin causes DNA breaks that are difficult to repair, hereby improving its tumor cell killing potential [22]. Oxaliplatin effectiveness is related to the expression level of nucleotide excision repair (*NER*) genes; indeed, ERCC1, XRCC1 and XDP, and WBSCR22 proteins represent novel oxaliplatin resistance biomarkers.

TGF-β1-treated CRC cells have been shown to increase epithelial mesenchymal transition, indicating the involvement of TGF-β1 in resistance to oxaliplatin [23,24,25]. 

Irinotecan (CPT-11), a semi-synthetic derivative of the plant extract camptothecin, is another chemotherapeutic drug used in CRC, that inhibits topoisomerase Ⅰ (Topo Ⅰ). In cells, Irinotecan becomes an active metabolite, SN-38, with a stronger anticancer activity, and forms a topoisomerase-inhibitor-DNA complex affecting the DNA function. Elevated levels of Topo I make cells more sensitive to irinotecan [26,27]. Furthermore, carboxylesterases (CES), uridine diphosphate glucuronosyltransferase (UGT), hepatic cytochrome P-450 enzymes CYP3A, β-glucuronidase and ATP-binding cassette (ABC) transporter protein, involved in the uptake and metabolism of Irinotecan, have a role in chemoresistance [28,29]. If the Irinotecan resistance is due to the epigenetic changes occurring in CRC, the use of histone deacetylase (HDAC) inhibitors could solve the resistance of the CRC cells to Irinotecan [30].

A second-line option for the combined treatment of mCRC (metastatic CRC), is represented by Capecitabine and Irinotecan therapy (XELIRI) with or without Bevacizumab [31,32]. The advent of monoclonal antibodies such as Bevacizumab and Cetuximab permitted great development in the CRC therapy.

In the last few years, studies have focused on stem cells and their prognostic value for CRC [33,34]. In fact, these cells show an enrichment of surface markers such as CD133, EphB2high, EpCAMhigh, CD44+, CD166+, ALDH+, LGR5+ and CD44v6+, which are useful for prognosis and follow the course of the pathology [35]; moreover, these cells show a higher expression of ATP binding cassette (ABC) family members, the efflux pumps that promote the transport of drugs outside the cell [36,37].

In addition to those described, other drugs in recent years have been used to treat CRC, including tyrosine kinase inhibitors (TKI) such as Sorafenib and Axitinib that block cell proliferation by inhibiting the mitogen-activated protein kinase (MAPK) pathway and prevent tumor-associated angiogenesis. Several studies have shown that the single use of TKIs is ineffective to increase patient survival and combined approaches are under investigation. Some clinical data suggested the use of Sorafenib in combination with oxaliplatin and irinotecan in metastatic CRC patients as it appears to block cell proliferation [38,39]; in other studies, sorafenib, used in combination with standard FOLFOX chemotherapy, was not effective [40,41,42]. Axitinib seems to have a better effect used in combination therapy with other chemotherapeutic drugs such as Erlotinib and Dasatinib [42].

Furthermore, Cisplatin is employed for the treatment of CRC, inducing the formation of platinum–DNA adducts [43], which in turn trigger the apoptotic process [44]. Cisplatin treatment often results in the development of resistance, leading to therapeutic failure. Intense research has identified several mechanisms underlying Cisplatin resistance [45,46].

Nutlin-3 is a chemotherapeutic drug that inhibits the interaction between Mouse double minute 2 homolog (MDM2) and tumor suppressor p53 causing the stabilization of p53 and its consequent activation. In this way p53 leads to the inhibition of cancer cell proliferation and the induction of cellular senescence. 

The Doxorubicin is an antineoplastic antibiotic of the anthracycline family with a broad antitumor spectrum. The drug binds to cellular DNA, inhibiting nucleic acid synthesis and mitosis and causing chromosomal aberrations. The literature shows that a combination of Nutlin-3 and Doxorubicin was more effective in treatment [47].

A novel important approach in cancer therapy is represented by the application of proteasome inhibitors [48,49,50,51]. In particular E3 ligases, enzymes that perform the final step in the ubiquitination cascade, represent drug targets for its ability to regulate protein stability and functions [52,53]. For this reason researchers are exploring the role of E3 ligases in tumor chemotherapy resistance and the underlying mechanism [54,55,56,57,58,59,60]. Indeed, a growing number of E3 ligases and related substrate proteins, such as the RING finger protein (RNFs), MDM2, the apoptotic protein inhibitor (IAPs), and tripartite proteins (TRIMs), have emerged as crucial players in drug resistance of several cancers, including CRC [61,62]. The literature reports that both C3HC4-typezinc finger-containing 1 (RBCK1), also known as HOIL-1L (a protein with an N-terminal ubiquitin like (UBL) domain) and the 3-ubiquitin ligase FBXW7 (a protein that influences the epithelial–stromal micro environmental interactions) increase epithelial-mesenchymal transition (EMT) and contribute to chemoresistance and stemness in CRC [63]. The miR-223/FBXW7 pathway has been reported to play a crucial role in the mechanism of chemoresistance in many human cancers, such as gastric, breast, and non-small cell lung cancers. However, it is unclear whether similar mechanisms of doxorubicin resistance are involved in particular in CRC. The miR-223/FBXW7 axis regulates doxorubicin sensitivity through EMT in CRC [64]. Moreover, the overexpression of RNF126, RING finger protein 126 (RNF126), a novel E3 ubiquitin ligase, was remarkably associated with multiple advanced clinical features of CRC patients independent of *p53* status. RNF126 promotes cell proliferation, mobility, and drug resistance in CRC via enhancing p53 ubiquitination and degradation [65]. Considering the resistance mechanisms described for the CRC, more research is needed to clarify the role in this mechanism of others E3 ligases such as TRIM proteins.

## 2. TRIM Proteins in Cancer 

The TRIpartite Motifs (TRIM) protein family is composed of more than 70 known TRIM proteins in humans and mice, which are encoded by approximately 71 genes in humans. The TRIpatite Motif is composed by three zinc-binding domains, a RING domain (R), a B-box type 1 (B1) and a B-box type 2 (B2), followed by a coiled-coil (CC) region [66,67]. Functionally, the RING finger domain is involved in the ubiquitination system, mediating the transfer of ubiquitin from E2-Ub ligase enzyme to its substrates: this domain is therefore a characteristic signature of many E3 ubiquitin ligases [68]. Genes encoding for TRIM proteins are present in all metazoans [69] and mutations in these genes are implicated in a variety of human diseases including cancer.

This is not surprising if we consider that TRIM family proteins are involved in a plethora of cellular functions, such as regulation of gene expression, signal transduction pathways, autophagy, cell growth, migration, protein stability through the ubiquitination system, regulation of development and immune response, effects on cell survival and metabolism and direct antiviral action. Alterations of TRIM expression levels represent biomarker and prognostic factors of specific cancers including osteosarcoma, gastric, liver, breast, ovarian, prostate, lung, cervical and CRC [70,71,72].

Depending on tumor type and on their deregulation mechanisms, TRIMs proteins can exert their action both as onco-protein and tumor-suppressor proteins in cancers. To date, many TRIMs proteins have resulted to be overexpressed in one or more cancers. TRIMs 11, 14, 22, 24, 25, 27, 28, 32, 37, 44, 47, 49, 59, 65 are upregulated in some of the high incidence cancers (breast, gastric, liver, lung, osteosarcoma, prostate, kidney) [73,74,75,76,77,78,79,80,81,82,83,84,85,86,87,88,89,90,91,92,93,94,95,96,97,98,99,100,101,102,103,104,105,106,107,108,109,110,111,112,113,114], while some others TRIM are upregulated in a cancer-specific way (e.g., TRIM22 in lung, TRIM31 and TRIM35 in liver, TRIM63 in breast, TRIM66 in osteosarcoma, TRIM68 in prostate). The altered expression of this TRIMs has been correlated with poor prognosis [115,116,117,118,119] (Table 1). On the contrary, there are also many TRIMs downregulated in tumors. TRIMs 3, 8, 13, 16, 21, 62 are downregulated in many of the main cancers worldwide (breast, gastric, liver, lung, osteosarcoma, prostate, kidney) [120,121,122,123,124,125,126,127,128,129,130,131], while some TRIMs are under-expressed in a cancer-specific way (e.g., TRIM15 in gastric cancer, TRIM26 in liver, TRIM58 in lung). In these cases, their downregulation is correlated with early-onset and poor overall survival among cancer patients [132,133,134] (Table 1).

Interestingly, there are some TRIM proteins, which result in being up- or down- regulated depending on cancer type. Among them are TRIM2 (up-regulated in osteosarcoma, down-regulated in kidney cancer), TRIM29 (up-regulated in lung cancer and osteosarcoma, down-regulated in liver and prostate cancers), TRIM33 (up-regulated in breast cancer, down-regulated in liver and kidney cancers) [135,136,137,138,139,140,141,142,143]. 

At the basis of the correlation between TRIMs, altered expression and tumor onset, there are, generally, several mechanisms, not fully understood, such as chromosomal translocations (resulting in oncogenic gain-of-function fusion genes), that likely contribute to oncogenesis through the constitutive activation of oncogenic signaling pathways, hyper- or hypo- methylation of CpG islands present in the TRIMs promoter regions [70,152,153,154,155,156]. Alternatively, the low expression of the tumor-suppressive TRIMs is inversely correlated with specific micro RNAs (miRs) overexpression (e.g., TRIM8 vs. miR17-92 family). On the contrary, overexpression of some oncogenic TRIMs in various cancers is frequently due to the loss of miR dependent gene suppression (e.g., *TRIM11, TRIM14, TRIM24, TRIM25, and TRIM44*) [144,150,157,158,159,160,161,162,163].

The pivotal role of TRIMs, in the pathological as well as in the physiological cellular life, is now clear if we consider that one or more TRIM members can influence diverse key downstream effector cellular pathways, such as p53 controlled pathways, the Wnt/β-catenin signaling, Transforming Growth Factor-β (TGF-β), Phosphoinositide-3-kinase /Protein Kinase B (PI3K/Akt) pathways and the pro-inflammatory Signal transducer and activator of transcription 3- Nuclear Factor kappa-light-chain-enhancer of activated B cells (STAT3-NF-κB) pathways.

### 2.1. TRIM Family and the p53 Controlled Pathways

The tumor suppressor protein p53 is a key player in the regulation of cell cycle, apoptosis, and in the maintenance of genome stability. A high percentage of tumors show inactivation of p53 function due to gene mutation (with a consequent not-functional p53 protein) or to network inactivation (also in the presence of a wild-type p53 protein). In several human malignancies, it has been shown that TRIMs are able to modulate chemoresistance by exerting their role on p53 stability and/or activity [164,165].

The oncogenic TRIMs are able to negatively regulate p53 by increasing its polyubiquitination and subsequent proteasomal degradation or by impairing its transcriptional activity both directly and indirectly (e.g., TRIM11, TRIM21, TRIM23, TRIM24, TRIM25, TRIM28, TRIM29, TRIM31, TRIM32, TRIM39, TRIM59 and TRIM66) [149,166,167,168,169,170,171,172,173,174,175,176,177,178,179]. On the contrary, a group of tumor suppressive TRIMs are showed to have positive stabilizing effects on p53 protein. They mainly work by enhancing p53 stability by interfering with MDM2 ubiquitin ligase activity and/or inhibiting the p53–MDM2 interaction. These TRIMs are downregulated in tumors (e.g., TRIM3, TRIM8, TRIM13, TRIM19 and TRIM67) [145,150,151,180,181,182,183,184] (Table 1).

### 2.2. TRIM Family and the Wnt/β -Catenin Signaling Pathway

The Wnt signaling pathway is involved in controlling several main cellular processes (e.g., proliferation, migration, cell adhesion). Moreover, it is important for normal embryonic development and adult tissue homeostasis [185,186].

TRIM29 and TRIM58 are the only two TRIM family members which have been identified to act by modulating the Wnt/β-catenin signaling pathway, and they act in an opposite way. TRIM29 (pro-proliferative) is upregulated in several human tumors. It induces the Wnt/β-catenin signaling pathway through upregulation of CD44 expression, linking this network to the progression of other human tumors. On the contrary, TRIM58 (anti-proliferative) is downregulated in human lung tumor. It exerts its tumor-suppressive activity by suppressing the expression of EMT and matrix metalloproteinase (*MMP*) genes and, consequently, by inhibiting cell invasion. Moreover, its overexpression significantly increases β-catenin ubiquitination and proteasomal degradation in gastrointestinal (GI) tumors [147,187,188,189,190,191,192].

### 2.3. TRIM Family and the TGF-β Pathway

Several members of TRIM proteins are implicated in the regulation of TGF-β signaling. The members of the TGF-β family are cytokines crucially involved in the regulation of cellular processes (e.g., cell growth, differentiation, migration, autophagy, and apoptosis). The TRIM family proteins work by specifically degrading signaling modules involved in this pathway (TGF-β-receptors, R-Smads, and Co-Smads) [193,194,195,196,197].

Different TRIM proteins have been demonstrated to modulate the canonical TGF-β-Smad signaling pathway both positively and negatively, depending on the TRIM protein involved. In particular, TRIM14, TRIM25, TRIM27, TRIM44, TRIM47, TRIM59 are significantly linked to the TGF-β signaling pathway [198,199,200].

### 2.4. TRIM Family and the PI3K/Akt Signaling Pathway

The PI3K/Akt pathway has also been correlated with tumor onset and progression. Its deregulation is frequently observed in most human malignancies due to the altered transmission of extracellular growth factor-derived signals [201,202,203,204]. The activation of the PI3K/Akt pathway has been frequently observed in various cancers, but only for a few years has this activation been also linked to an increased expression of some TRIM proteins.

For example, TRIM14, TRIM27, TRIM44 and TRIM59 are linked to the activation of the PI3K/Akt pathway in different tumors. These TRIM members are upregulated in cancer tissues and this correlates with a poor prognosis and an increase in some characteristic tumor features including invasion, metastasis, and apoptosis resistance [205,206,207,208,209]. They act by degrading its antagonist Phosphatase and tensin homolog (PTEN), with the increase in Akt phosphorylation and PI3K/Akt signaling activity, or by directly inducing the Akt signaling pathway, through the regulation of phosphorylated PI3K and Akt levels. Both these mechanisms lead to an increase in cell proliferation and EMT, with the consequent poor patients’ prognosis [144,210,211].

### 2.5. TRIM Family and the Pro-Inflammatory STAT3-NF-κB Pathway

Many tumors show a constitutive activation of transcription factors involved in the pro-inflammatory response. These include transcription factors like STAT3 and members of the NF-κB protein family, that seem to be crucial in linking chronic inflammation to cancer development. In particular, the aberrant STAT3 signaling is mainly due to persisting signaling events caused by the deregulation of specific signaling modules [212,213,214,215,216,217,218]. The aberrant TRIMs expression seems to be clinically relevant for constitutive STAT signaling in several tumors, particularly those of the gastrointestinal (GI) tract. Indeed, excessive TRIM-mediated *STAT3* activation has been reported for several TRIMs (e.g., TRIM14, TRIM27, TRIM29 and TRIM52) and this is associated with an overall poor survival of patients [219,220,221,222]. Oncogenic TRIMs, in particular, exert their effects mainly through the activation of the Janus kinase/signal transducers and activators of transcription 3 (JAK/STAT3) signaling pathway by inducing the formation of the constitutively active JAK1-2/STAT3 complex or by promoting the poly-ubiquitination and consequent degradation of a protein tyrosine phosphatase involved in the negative regulation of *STAT3* (named Shp2), thus activating a STAT3 signal. By promoting the activation of *STAT3*, TRIMs indirectly induce STAT3-target genes, such as *MMP-2*, *MMP-9* and the vascular endothelial derived growth factor (VEGF), thus promoting cancer cell migration and invasion [220].

Moreover, the aberrant NF-κB activation is linked with several tumors’ onset. The NF-kB canonical pathway can be mainly activated by proteasomal degrading the NF-κB inhibitor IκBα, leading to a release and subsequent activation of dimeric complexes of the NF-κB/Rel transcription family members p50, p65 (RelA) and c-Rel. Alternatively, the NF–κB activation relies on the inducible phosphorylation–dependent ubiquitination and processing of the NF–κB precursor protein p100 by the action of the NF–κB-inducing kinase (NIK) (non-canonical pathway). Once activated, NF-κB promotes tumorigenesis by inducing proinflammatory genes such as cyclooxygenase-2 (*COX-2*).

Several oncogenic TRIMs are able to activate both the NF–κB dependent routes: the canonical NF-κB pathway, triggered by different pro-inflammatory cytokines, or the non-canonical NF-κB pathway, also by interfering with autophagy [223,224,225,226,227].

Contrary to oncogenic TRIMs, there are also some tumor suppressors, TRIMs protein that are able to antagonize NF-kB activity by promoting inhibitor of nuclear factor kappaB kinase subunit gamma (IKKγ) neddylation with the consequent stabilization of the IκBα protein and the impairing of the NF–κB activation, even in the presence of NF-κB activating cytokines [228].

## 3. TRIMs Involved in CRC 

Different TRIM proteins are involved in development and progression of CRC. They can regulate various aspects of tumorigenesis, including proliferation, apoptosis, autophagy, transcriptional regulation, chromatin remodeling, invasion, metastasis and chemoresistance [229]. Cancer cells, through different mechanisms such as inactivation of the tumor suppressor gene *p53*, may acquire resistance to chemotherapy. For this reason, the reactivation of wild type *p53*, through the TRIM proteins, could be a promising strategy to restore sensitivity to the treatment of chemotherapy in all tumors including CRC [164,165]. Below we describe the role and the levels of the different TRIMs in the CRC.

TRIM23 is upregulated in CRC, it binds p53, inducing its ubiquitination and promoting colorectal cell proliferation [149]; TRIM24 (transcription intermediary factor 1α-TIF1α), mRNA and protein levels were higher in CRC tissues compared to controls, indicating this TRIM is a potential negative prognostic marker. In particular, TRIM24 promotes the degradation of p53 via ubiquitination [230]. 

The literature reports that TRIM25 negatively regulates the expression of Caspase-2, and consequently the reduction of TRIM25 levels in the colorectal cell increases their sensibility to drugs [230]; moreover, TRIM25 reduction induces p53 acetylation and p53-dependent cell death in HCT116 cells [173,231,232]. This TRIM regulates p53 levels and activity in the HCT116 cell line in two opposite ways. From one side, TRIM25 prevents the formation of the ternary complex constituted by p53, MDM2, and p300, which is essential for p53-polyubiquitination and degradation, leading to the increase in p53 stability. Despite this, from another side, p53 transcriptional activity is inhibited in the presence of TRIM25, since the same p53-MDM2-p300 complex is required for p53 acetylation and, consequently, it is able to block the p53-dependent activation of p53-controlled apoptotic genes, following DNA damage [173]. 

TRIM59 is upregulated in CRC patients and correlates with a poor prognosis. Therefore, the reduction of TRIM59 levels reversed the expression of epithelial-mesenchymal transformation-related proteins vimentin, in *p53* wild-type and *p53* mutated cells, demonstrating that the TRIM59 oncogenic action is *p53* independent [209,233]. In CRC, it has not been studied whether the oncogenic role of TRIM59 is through direct degradation of p53; only in stomach cancer does the literature report a direct TRIM59-p53 interaction and subsequent p53 degradation [170].

TRIM28 and TRIM29 are markers for patient survival in CRC. TRIM28 binds MDM2 and promotes the degradation of p53 [234]. In addition, TRIM28, in concert with MDM2, promotes the formation of a p53 complex with histone deacetylase 1 (HDAC1), thus preventing acetylation of p53 [166]. To date, it is unclear by which molecular mechanism TRIM29 regulates the development of CRC; in fact, this TRIM could prevent p53-mediated transcription of its target genes in the nucleus by sequestering p53 outside the nucleus and thus preventing its p300-dependent acetylation [168]. Alternatively, it could promote p53 degradation by degrading and/or changing the localization of TIP60, a transcriptional coactivator of p53, consequently reducing TIP60-dependent p53 acetylation [146]. In contrast, histone deacetylase9 (HDAC9) can inhibit the action of TRIM29, resulting in increased p53 activity and reduced cell survival [168].

Some TRIM proteins behave like oncogenes because they are involved in the activation of pro-proliferative pathways such as Akt/mTOR and NF-κB signaling pathways. In particular, TRIM2 and TRIM47 are potential targets for therapy in CRC, since they promote cell proliferation, epithelial-mesenchymal transition (EMT) and metastasis in vitro and in vivo [143,199]. TRIM6 is upregulated in CRC and its reduction increases the anti-proliferative effects of 5-fluorouracil and oxaliplatin [235]. TRIM27 and TRIM44 are involved in activation of the Akt/mTOR signaling pathway inducing cell proliferation, migration, invasion and metastasis in CRC [206,207]. Additionally, TRIM66, TRIM52 and TRIM14 also play an oncogenic role in CRC, since they are involved in cancer proliferation and metastasis through the regulation of STAT3 pathway expression [118,144,220,222]. Instead, TRIM27 is involved in proliferation, invasion and metastasis of CRC in vitro and in vivo regulating AKT [206]. TRIM14, together with TRIM1, have a role in autophagy. Indeed, TRIM14 negatively interferes with the autophagic degradation of the NF-κB family member p100/p52, inducing a non-canonical NF-κB signaling pathway [227]. By contrast, TRIM11 mediates the degradation of the receptor-interacting protein kinase 3 (RIPK3). RIPK3 activation is linked to necrotic cell death and represents a causative role for both pediatric and adult IBDs (inflammatory bowel diseases). TRIM11 counteracts mTOR-induced activation of RIPK3, inducing RIPK3 degradation through autophagy and thus representing a novel regulatory mechanism important for antagonizing necroptosis [236]. In this way, TRIM11 shows a protective role in the gut, mainly through antagonizing intestinal inflammation and cancer. 

Finally, TRIM31 and TRIM40 interfere with the canonical NF–κB pathway, promoting invasion and metastasis in CRC [226,227]. In particular, TRIM40 is downregulated in gastrointestinal cancers. It is able to inhibit the NF-κB activity by promoting the neddylation of the IKKγ, also called NEMO (NF–κB essential modulator), a key regulator for NF-κB activation, thus preventing inflammation-associated carcinogenesis in the GI tract [202]. In contrast to the oncogenic action of the TRIMs described so far, TRIM67, TRIM58 and TRIM8 are downregulated in CRC, playing a tumor suppressor role. The reduction of TRIM67 levels in CRC is caused by methylation of two loci (cg21178978 and cg27504802). Mechanistically, TRIM67 binds p53, thus inhibiting MDM2 binding to p53 and following ubiquitination [184]. TRIM58 plays a critical role of tumor suppressor by limiting Wnt/β-catenin dependent EMT; indeed, the recovery of TRIM58 reduces tumor invasion [191]. TRIM8 seems to be down-regulated in CRC and in restoring TRIM8 levels, p53 is stabilized, and cells become sensitive again to chemotherapeutics (The Human protein Atlas, available from http://www.proteinatlas.org, accessed on 25 February 2021) [151,182,229,237] (Table 2).

## 4. *TRIM8* Tumor Suppressor Gene

The *TRIM8* gene is located on the 10q24.3 chromosome and transcribes an mRNA of about 3.0 kbp that is translated into a protein of 551 aa with a molecular weight of 61.5 kDa. TRIM8 is expressed in many human tissues such as lungs, intestine, breast, brain, placenta, muscles, kidneys. TRIM8 performs activities involved in embryonic development and cell differentiation, in response to the innate immune system and in different human tumors [238,239]. Although a role of *TRIM8* as an oncogene is reported by affecting the NF-κB and JAK-STAT pathways, much experimental evidence support a role for *TRIM8* as a tumor suppressor [240,241,242,243]. Over the years it has been demonstrated how these two pathways are involved in several processes, including inflammatory ones that are associated with the onset and development of CRC [244,245,246]. 

The first role of TRIM8 in cancer was demonstrated by Vincent et al., in 2000. The authors showed frequent deletion or loss of heterozygosity in the *TRIM8* gene in glioblastomas [238]. Then, Carinci et al. performed the transcriptome analysis of larynx squamous cell carcinoma (LSCC) tissue and they observed a large reduction of TRIM8 expression which correlated with metastatic progression, suggesting a tumor suppressor role of TRIM8 [148]. Over the years, the suppression role of TRIM8 has been observed in different tumors. Zelin et al. demonstrated that TRIM8 is downregulated in breast cancer and the protein level of TRIM8 is negatively correlated with estrogen receptor α. Moreover, knockdown of TRIM8 can significantly enhance breast cancer cell proliferation and migration both in vitro and in vivo [131]. As in breast cancer, and also in other cancers, low expression levels of TRIM8 are associated with a poor prognosis for the patients; in fact, TRIM8 is downregulated in Glioma, Chronic lymphocytic leukemia (CLL), Renal Clear Cell Carcinoma (ccRCC), CRC and in melanoma [148,150,247,248,249]. 

In particular, the literature reports that the downregulation of TRIM8 in tumors is often caused by the action of specific miRNAs. In fact, in patients affected by ccRCC, CRC (The Human protein Atlas, available from http://www.proteinatlas.org, accessed on 25 February 2021) [237], Glioma, and CLL, the overexpression of miR-17-5p causes TRIM8 downregulation that affects cell proliferation and is associated with patient’s survival [150,229,247,248]. One of the reasons why TRIM8 plays a tumor suppressor role is its capacity to regulate the stability and activity of *p53* tumor suppressor gene. Indeed, *TRIM8* is a direct p53 target gene, and by a feedback mechanism, displaces p53-MDM2 binding, thus stabilizing p53 and promoting MDM2 degradation. As final outcome, TRIM8 promotes the p53-dependent suppression of cell proliferation and DNA repair [182]. 

Generally, the most aggressive chemo-resistant tumors have mutations in the *p53* gene or inactivation in its pathway through alterations of its regulators. This is the case with tumors like the clear cell renal cell carcinoma (ccRCC) and the CRC in which the reactivation of the p53 pathway could be one of the best treatment strategies [250,251,252]. Strikingly, it has been demonstrated that in HCT116 colon, carcinoma and in ccRCC cell lines, TRIM8 silencing induced p53 inactivation and MDM2 stabilization impairing Cisplatin and Nutlin-3 effect. This suggests that TRIM8 levels are relevant to the p53-mediated cellular responses to chemotherapeutic drugs. Conversely, the overexpression of TRIM8 in HCT116 cells induced a great reduction in proliferation rate, which became more pronounced when the cells were treated with Nutlin-3 and Cisplatin [150]. 

Another case of resistance to chemotherapy is represented by Anaplastic Thyroid Cancer (ATC), where it has been demonstrated that TRIM8 is a direct target of miR-182, which is upregulated in ATC tissue and cell lines. Suppressing the action of TRIM8, miR-182 promotes cellular growth and enhances the cisplatin resistance of ATC cells [253].

## 5. TRIM8 and miR-17-92 Cluster in CRC Progression and Chemo Resistance

The downregulation of TRIM8 expression in tumors, including CRC, is explained by the upregulation of the miR-17-5p belonging to the miR-17-92 cluster. MiR-17-5p directly targets the 3′ UTR of TRIM8 repressing its expression [150,247,248]. The human genome contains two paralogues of the miR-17-92 cluster, the miR-106b/25 cluster and the miR-106a/363 cluster. The miR-17-92 and miR-106b/25 clusters are emerging as key actors in a wide range of biological processes including tumorigenesis [254,255]. An increasing number of recent papers has reported that miR-106b- 5p and miR-17-5p are overexpressed in many different chemo/ radio-resistant cancers, including CRC, ccRCC and glioma, playing a role in early metastatic progression [256] and contributing to oncogenesis and chemo-resistance [150]. MiR-17-5p and miR-106b-5p are transactivated by N-MYC, and this oncogene is negatively regulated by miR-34a, which is transactivated by p53. Interestingly, miR-17-5p and miR-106b-5p silencing increases TRIM8 expression levels, which in turn stabilizes and activates p53 towards a cell proliferation arrest program. Moreover, p53 promotes the transcription of miR-34a, which turns off the oncogenic effect of N-MYC, linking p53 to N-MYC. By restoring normal TRIM8 levels, CRC cells recover sensitivity to chemotherapy treatments such as Sorafenib, Axitinib, which are among the Tyrosine Kinase inhibitors currently in use for treatment of both renal and colorectal carcinoma [257,258,259,260], and also to Nutlin-3 and Cisplatin [150]. In conclusion, TRIM8, among all miR-17-5p targets, is pivotal in controlling cell sensitivity to chemotherapy and its role in tumor growth has been demonstrated also in human tumor xenografts generated in nude mice. Indeed, in TRIM8-treated tumors, cell proliferation stops completely compared to tumors treated with a control vector. This evidence confirmed in vivo the pathway identified in vitro, underlying TRIM8 as a key factor in the p53/N-MYC/miR-17 axis [150].

Another important role of TRIM8 in counteracting the proliferation of cancer cells is highlighted by its effects on the stability and activity of the oncogenic transcription factor ΔNp63α, belonging to the *p53* gene family. ΔNp63α is upregulated in different tumors, in fact the expression level of this transcription factor is correlated with a poor prognosis of patients [261,262,263]. It has been demonstrated that TRIM8 promotes the degradation of ΔNp63α in both a proteasomal and caspase-1-dependent way. It is important to point out that ΔNp63α is able to downregulate TRIM8 expression, thus preventing the stabilization of p53. This dual role of TRIM8 demonstrates an enhanced activity in the inhibition of tumor development and therefore in the role played in chemoresistance and offers more possible therapeutic benefits [264].

## 6. Conclusions

Studies are increasingly focusing on the molecular basis of chemoresistance in CRC. The identification of new molecular targets and the development of drugs able to regulate their activity opens a positive landscape for CRC patients. Specifically, in this review we reported the tumor suppressor role of TRIM8 in the resistance to drugs administered for the treatment of CRC. TRIM8 is downregulated in colon carcinoma cells due to the inhibitory action of miR-17-5p and miR-106b. Suppressing the activity of these miRNAs, the level of TRIM8 proteins increase, and the activity of p53 tumor suppressor protein is restored and cells respond again to chemotherapy treatment. TRIM8 is therefore a promising therapeutic target for CRC treatment.

## Figures and Tables

**Table 1 biomedicines-09-00241-t001:** TRIM proteins in tumors. Tripartite motif (TRIM) proteins described in the manuscript are listed based on their expression levels in the main cancer types worldwide.

Cancer Type	Upregulated TRIMs	Downregulated TRIMs
**Breast**	**11** [112], **24** [77], **25** [73], **27** [75], **28** [74], **32** [107], **33** [138], **37** [82], **44** [97], **47** [113], **59** [104], **63** [119]	**8** [131], **13** [130], **16** [127], **21** [128], **62** [120]
**Gastric**	**14** [144], **24** [84], **28** [76], **32** [96], **37** [106], **44** [79], **59** [105]	**3** [145], **15** [134]
**Liver**	**11** [92], **14** [101], **24** [108], **28** [90], **31** [117], **32** [85], **35** [116], **37** [83], **65** [100]	**3** [121], **16** [125], **21** [123], **26** [132], **29** [146], **33** [135]
**Lung**	**11** [109], **22** [98], **24** [80], **25** [88], **27** [78], **28** [81], **29** [139], **32** [111], **37** [102], **44** [91], **47** [93], **59** [94], **65** [89]	**13** [129], **16** [124], **58** [133], **62** [122]
**Osteosarcoma**	**2** [142,143], **29** [147], **37** [99], **59** [87], **66** [118]	**8** [148]
**Prostate**	**24** [110], **25** [149], **28** [103], **47** [86], **68** [115]	**16** [126], **29** [136]
**Renal**	**44** [114], **59** [95]	**2** [141], **8** [150,151], **33** [137,140]

**Table 2 biomedicines-09-00241-t002:** TRIM proteins involved in CRC. Tripartite motif (TRIM) proteins described in the manuscript are listed based on their expression levels in different types of cancer, also below is indicated where the TRIM gene expression levels or mechanisms of action were obtained. The arrows indicate if that TRIM protein was found up- (↑) or down- (↓) regulated; nd indicates not detected levels.

TRIMs	Levels	Action		References
**TRIM11**	nd	Has a protective role in the gut mainly through antagonizing intestinal inflammation and cancer	HEK293 and HT29 cell line	[236]
**TRIM14**	↑	promotes migration and invasion of CRC regulating the SPHK1/STAT3 pathway	CRC tissue and HT-29, SW620, and LoVo cell lines	[220]
**TRIM2**	↑	promotes Epithelial-mesenchymal transition in CRC	CRC tissue and SW620, RKO cell lines	[143]
**TRIM23**	↑	induces p53 ubiquitination promoting cell proliferation	CRC tissue, SW480, HT29, SW1116, HCT116, SW620 and FHC cell lines, xenograft	[179]
**TRIM24**	↑	mRNA and protein levels are elevated in CRC tissue	CRC tissue	[230]
**TRIM25**	↑	its reduction increases the CRC chemo sensibility	DLD-1RKO and HEK293 cell lines	[231]
**TRIM27**	↑	involved in proliferation, invasion and metastasis of CRC	CRC tissue, LoVo, HCT116, SW480, DLD-1, HT29 and normal epithelial colon cells (NCM460), xenograft	[206]
**TRIM28**	↑	promotes MDM2-mediated p53 degradation reducing the CRC patient survival	CRC tissue	[234]
**TRIM29**	↑	prevents p53-mediated transcription of its target genes	CRC tissue, CT116, SW620, SW480, SW1116, LOVO, HT29 and RKO cell lines	[219]
**TRIM31**	↑	involves in canonical NF–κB pathway, promotes invasion and metastasis in CRC	CRC tissue, HT-29, SW 116, SW 620, SW 480 cell lines	[226]
**TRIM40**	↑	involves in activation of the Akt/mTOR signaling pathway, inducing cell proliferation, migration, and invasion in CRC	CRC tissue, HEK293T, HeLa and SW480 cell lines	[228]
**TRIM44**	↑	involves in activation of the Akt/mTOR signaling pathway, induces cell proliferation, migration, and invasion in CRC	CRC tissue, Intestinal mucosal epithelial cells (NCM460) and SW620, LOVO, and HCT116 cell lines	[207]
**TRIM47**	↑	promotes proliferation and metastasis in CRC	CRC tissue, HCT116, HT29, SW480, RKO, SW620, Caco2, LoVo and SW1116 cell lines, nude mice	[200]
**TRIM52**	↑	with an oncogenic role in CRC via regulating the STAT3 signaling pathway	CRC tissue, SW480, LoVo, SW620, HT29 and RKO) and normal human intestinal crypt cells (HIEC), xenografts	[222]
**TRIM59**	↑	the reduction of TRIM59 levels reduce the expression of EMT related proteins	CRC tissue, Caco-2, SW480, HT-29, LoVo, DLD-1, HCT116 cell lines and normal human colorectal epithelial cells (NCM460)	[208]
**TRIM6**	↑	its reduction increases the anti-proliferative effects of 5-fluorouracil and oxaliplatin	CRC tissue, FHC, and CRC cell lines, LOVO, Sw620, HCT-8 and HCT116 cell lines and nude mice	[235]
**TRIM66**	↑	regulates migration and invasion in CRC through JAK2/STAT3 pathway	CRC tissue, Human normal colorectal cell lines NCM460 and human CRC cell lines including HCT116, HT29, CaCo2 and SW620	[118]
**TRIM58**	↓	inhibits CRC invasion through EMT and MMP activation.	CRC tissue, HCT8, KM12, Caco-2, DLD-1, HCT116, LoVo, HT-29, SW480, SW620, RKO and HCT15 cell lines	[191]
**TRIM67**	↓	inhibits metastasis by mediating mitogen-activated protein kinase 11 (MAPK11) in CRC	CRC tissue and xenografts	[184]
**TRIM8**	↓	restoring levels of TRIM8 the CRC becomes sensitive to chemotherapeutic drugs	HCT116 cell line and xenografts	[151,182]

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
