# Peer review of "TRIM Proteins in Colorectal Cancer: TRIM8 as a Promising Therapeutic Target in Chemo Resistance"

_biomedicines, 2021, doi:10.3390/biomedicines9030241_

Round 1
Reviewer 1 Report
It is well written review paper about TRIM proteins in colorectal cancer.
Minor point;
Is there any relationship between TRIM family and well known predictive or prognostic marker of colorectal cancer, such as tumor location, lymph node metastasis, tumor budding, RAS/RAF mutation, serum CEA, etc.
Author Response
Dear Editor,
Thank you for considering our manuscript (Manuscript ID:biomedicines-1093518): “TRIM proteins in Colorectal Cancer: TRIM8 as a promising therapeutic target in chemo resistance” by Flaviana Marzano et al. for publication in Biomedicines-Special Issue "Colorectal Cancer: From Pathophysiology to Novel Therapeutic Approaches”.
We sincerely thank you and the reviewers for constructive criticism and valuable comments, which were of great help in revising the manuscript.
We have carefully considered the reviewer's comments and address their points and fullfill their requests.
The revised version of the manuscript contains all the revisions required with new tables and the revised version.
You can see the changes made by following the revisions indicated in the text.
Your Sincerely,
Apollonia Tullo
REVIEWER 1
“Is there any relationship between TRIM family and well known predictive or prognostic marker of colorectal cancer, such as tumor location, lymph node metastasis, tumor budding, RAS/RAF mutation, serum CEA, etc.”
Answer
- We thank the reviewer for this comment. We reported several times in the text that TRIM proteins are considered potential biomarkers for CRC. Indeed, TRIM proteins deregulation in patients affected by CRC correlated with poor prognosis and an increase in some tumor features including invasion, metastasis, and apoptosis resistance. No correlation between TRIM proteins and CRC markers such as tumor location, tumor budding, RAS/RAF mutation or serum CEA has been reported in the literature.

Reviewer 2 Report
There is no doubt that the authors have made a substantial effort to cover a large part (if not all) of the literature dealing with the topic of this Review. Thus, the manuscript contains many results, from many studies, which sometimes difficult to summarize and integrate. However, it is clear from the elevated number of paragraphs constituted by single sentences, that the manuscript needs more work. One sentence cannot form a paragraph and the authors would need to integrate all the information gathered in a story that flows. One possibility could be to prepare more tables and some schemes.
During the manuscript, the authors refer several times to the Human Protein Atlas. To the best of my knowledge, this is a website that gathers information obtained by researchers. If this is so, the authors would need to reference the original studies rather than the webpage.
It is critical to specify if the result included, particularly those related to gene expression levels or mechanisms of action, were obtained in commercial cell lines, xenografts, organoids, or using human tissues.
Finally, even though this is a review, it would be really interesting if the authors could do a simple bioinformatic approach, investigating the expression of TRIM proteins in human CRC biopsies in datasets publicly available (in NCBI GEO or The Cancer Genome Atlas), and particularly, if there is a relationship between expression levels and the Consensus Molecular Subtypes for CRC recently defined by Guiney et al (Nat. Med. 2015, 21, 1350–1356).
Author Response
Dear Editor,
Thank you for considering our manuscript (Manuscript ID:biomedicines-1093518): “TRIM proteins in Colorectal Cancer: TRIM8 as a promising therapeutic target in chemo resistance” by Flaviana Marzano et al. for publication in Biomedicines-Special Issue "Colorectal Cancer: From Pathophysiology to Novel Therapeutic Approaches”.
We sincerely thank you and the reviewers for constructive criticism and valuable comments, which were of great help in revising the manuscript.
We have carefully considered the reviewer's comments and address their points and fullfill their requests.
The revised version of the manuscript contains all the revisions required with new tables and the revised version.
You can see the changes made by following the revisions indicated in the text.
Your Sincerely,
Apollonia Tullo
REVIEWER 2
We really thank the reviewer for the insightful comments and great suggestions. We have carefully evaluated them to address the points raised and fulfill the requests.
All the revisions in the manuscript are in red. Below our answers and clarifications point-by-point
- There is no doubt that the authors have made a substantial effort to cover a large part (if not all) of the literature dealing with the topic of this Review. Thus, the manuscript contains many results, from many studies, which sometimes difficult to summarize and integrate. However, it is clear from the elevated number of paragraphs constituted by single sentences, that the manuscript needs more work. One sentence cannot form a paragraph and the authors would need to integrate all the information gathered in a story that flows. One possibility could be to prepare more tables and some schemes.
Answer
We fully agree with the reviewer that the manuscript contains many studies since TRIM proteins are numerous and involved in many processes and pathways. We modified the text trying to make it more fluent and clearer. Moreover we added a new table that report all the TRIM proteins involved in cancer (Table 1 in paragraph “TRIM Proteins in cancer”) to make the test even clearer.
- During the manuscript, the authors refer several times to the Human Protein Atlas. To the best of my knowledge, this is a website that gathers information obtained by researchers. If this is so, the authors would need to reference the original studies rather than the webpage.
Answer
We thank the reviewer for pointing out this oversight. We added the reference, which refers to human Protein atlas. Lines 383-423.
- It is critical to specify if the result included, particularly those related to gene expression levels or mechanisms of action, were obtained in commercial cell lines, xenografts, organoids, or using human tissues.
Answer
We really thank the reviewer for this evaluable suggestion that improved our paper.
We reported in Table 2 the relative information where the TRIM gene expression levels or mechanisms of action were obtained (in cells, tissues, or mice).
- Finally, even though this is a review, it would be really interesting if the authors could do a simple bioinformatic approach, investigating the expression of TRIM proteins in human CRC biopsies in datasets publicly available (in NCBI GEO or The Cancer Genome Atlas), and particularly, if there is a relationship between expression levels and the Consensus Molecular Subtypes for CRC recently defined by Guiney et al (Nat. Med. 2015, 21, 1350–1356).
Answer
As suggested by the reviewer we analysed the relationship between TRIM protein expression and the Consensus Molecular Subtypes (CMS) for CRC recently defined by Guiney et al. and deposited at link: doi:10.7303/syn2623706. In these data, which are the results they obtained, we identified only TRIM13 whose expression is not altered and therefore is unrelated to any CMS. It could be interesting to investigate for TRIM proteins in the initial data, but it requires a research activity that could be the topic for a research paper and not a review.

Round 2
Reviewer 2 Report
The manuscript has improved substantially, although there are still numerous single-sentence paragraphs.